**Cite this article:** Šubelj L, Waltman L, Traag V, van Eck NJ. 2020 Intermediacy of publications. *R. Soc. open sci.* **7**: 190207.

computational physics/complexity/graph theory

intermediacy, publication, citation network, main path analysis

**Author for correspondence:**
Lovro Šubelj
e-mail: lovro.subelj@fri.uni-lj.si

# Intermediacy of publications

Lovro Šubelj[1], Ludo Waltman[2], Vincent Traag[2]
and Nees Jan van Eck[2]

[1]University of Ljubljana, Faculty of Computer and Information Science, Ljubljana, Slovenia
[2]Leiden University, Centre for Science and Technology Studies, Leiden, The Netherlands

LŠ, 0000-0002-4161-2260

Citation networks of scientific publications offer fundamental insights into the structure and development of scientific knowledge. We propose a new measure, called intermediacy, for tracing the historical development of scientific knowledge. Given two publications, an older and a more recent one, intermediacy identifies publications that seem to play a major role in the historical development from the older to the more recent publication. The identified publications are important in connecting the older and the more recent publication in the citation network. After providing a formal definition of intermediacy, we study its mathematical properties. We then present two empirical case studies, one tracing historical developments at the interface between the community detection literature and the scientometric literature and one examining the development of the literature on peer review. We show both conceptually and empirically how intermediacy differs from main path analysis, which is the most popular approach for tracing historical developments in citation networks. Main path analysis tends to favour longer paths over shorter ones, whereas intermediacy has the opposite tendency. Compared to the main path analysis, we conclude that intermediacy offers a more principled approach for tracing the historical development of scientific knowledge.

## 1. Introduction

Citation networks provide invaluable information for tracing historical developments in science. The idea of tracing scientific developments based on citation data goes back to Eugene Garfield, the founder of the Science Citation Index. In a report published more than 50 years ago, Garfield *et al.* [1] concluded that citation analysis is 'a valid and valuable means of creating accurate historical descriptions of scientific fields'. Garfield also developed a software tool called HistCite that visualizes citation networks of scientific publications. This tool supports users in tracing historical developments in science, a process sometimes referred to as *algorithmic historiography* by Garfield et al. [2–4]. More recently, a software tool called CitNetExplorer [5] was developed that has similar functionality but offers more flexibility

in analysing large-scale citation networks. Other software tools, most notably CiteSpace [6] and CRExplorer [7,8], provide alternative approaches for tracing scientific developments based on citation data.

Main path analysis, originally proposed by Hummon & Doreian [9], is a widely used technique for tracing historical developments in science. Given a citation network, main path analysis identifies one or more paths in the network that are considered to represent the most important scientific developments. Many variants and extensions of main path analysis have been proposed [10–16], not only for citation networks of scientific publications but also for patent citation networks [17–21]. However, despite the large body of literature in which main path analysis is used, we question whether the technique is really suitable for tracing historical developments in science. We show that main path analysis has the tendency to favour longer citation paths over shorter ones. In our view, this is an undesirable property that leads to counterintuitive results.

As an alternative to main path analysis, we introduce a new approach for tracing historical developments in science based on citation networks. This approach is based on a measure that we call intermediacy. Given two publications dealing with a specific research topic, an older publication and a more recent one, intermediacy can be used to identify publications that appear to play a major role in the historical development from the older to the more recent publication. These are publications that, based on citation links, are important in connecting the older and the more recent publication.

Like main path analysis, intermediacy can be used to identify paths between publications in a citation network. However, as we show both conceptually and empirically, there are fundamental differences between intermediacy and main path analysis. Most significantly, whereas main path analysis tends to favour longer citation paths over shorter ones, intermediacy has the opposite tendency. For the purpose of tracing historical developments in science, we argue that intermediacy yields better results than main path analysis.

Intermediacy might seem similar to centrality, but there is an essential difference. Centrality measures [22], such as degree centrality, closeness centrality, betweenness centrality and eigenvector centrality, indicate how central a node is in a network. Intermediacy is different because it is defined relative to a specific source and target node, not relative to a network as a whole. This is why centrality measures cannot be used to capture the idea of intermediacy.

## 2. Intermediacy

Consider a directed acyclic graph $G = (V, E)$, where $V$ denotes the set of nodes of $G$ and $E$ denotes the set of edges of $G$. The edges are directed. We are interested in the connectivity between a source $s \in V$ and a target $t \in V$. Only nodes that are located on a path from source $s$ to target $t$ are of relevance. We refer to such a path as a source-target path. We assume that each node $v \in V$ is located on a source-target path.

**Definition 2.1.** Given a source $s$ and a target $t$, a path from $s$ to $t$ is called a *source-target path*.

In this paper, our focus is on citation networks of scientific publications. In this context, nodes are publications and edges are citations. We choose edges to be directed from a citing publication to a cited publication. Hence, edges point backward in time. This means that the source is a more recent publication and the target an older one.

Informally, the more important the role of a node $v \in V$ in connecting source $s$ to target $t$, the higher the intermediacy of $v$. To formally define intermediacy, we assume that each edge $e \in E$ is either *active* or *inactive*. An edge is active with a certain probability $p$, where $p \in (0, 1)$. This probability is the same for all edges. We exclude the possibility that this probability equals 0 or 1, since this would not yield useful results. Based on the notion of active and inactive edges, we introduce the following definitions.

**Definition 2.2.** If all edges on a path are active, the path is called *active*. Otherwise, the path is called *inactive*. If a node $v \in V$ is located on an active source-target path, the node is called *active*. Otherwise, the node is called *inactive*.

For two nodes $u, v \in V$, we use $X_{uv}$ to indicate whether there is an active path (or multiple active paths) from node $u$ to node $v$ ($X_{uv} = 1$) or not ($X_{uv} = 0$). The probability that there is an active path from node $u$ to node $v$ is denoted by $\Pr(X_{uv} = 1)$. We use $X_{st}(v)$ to indicate whether there is an active source-target path that goes through node $v$ ($X_{st}(v) = 1$) or not ($X_{st}(v) = 0$). The probability that there is an active source-target path that goes through node $v$ is denoted by $\Pr(X_{st}(v) = 1) = \Pr(X_{sv} = 1)\Pr(X_{vt} = 1)$. This probability equals the probability that node $v$ is active.

Intermediacy can now be defined as follows.

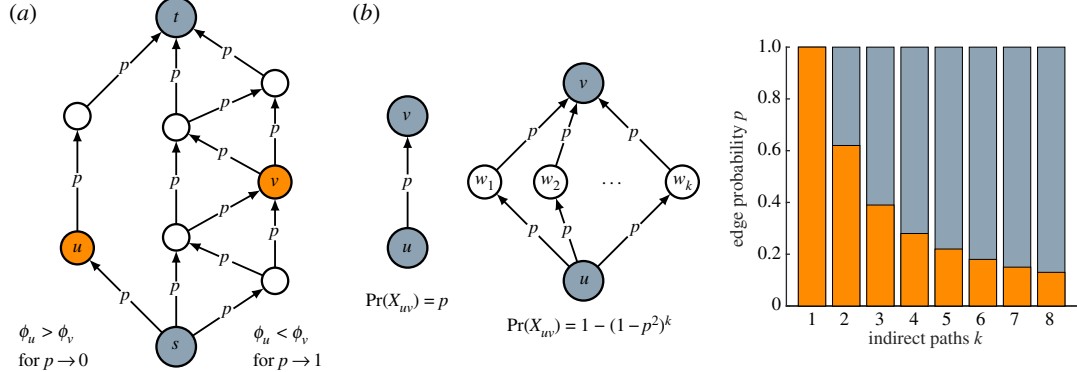

**Figure 1.** (a) Illustration of the limit behaviour of intermediacy. For $p \to 0$, intermediacy favours nodes located on shorter paths and therefore node $u$ has a higher intermediacy than node $v$. For $p \to 1$, intermediacy favours nodes located on a larger number of edge-independent paths and therefore node $v$ has a higher intermediacy than node $u$. (b) Illustration of the choice of the parameter $p$. Nodes $u$ and $v$ are connected by a single direct path in the left graph and by $k$ indirect paths of length 2 in the right graph. For different values of $k$, the bar chart shows the values of $p$ for which the probability that there is an active path from node $u$ to node $v$ is higher (in orange) or lower (in grey) in the left graph than in the right graph.

**Definition 2.3.** The *intermediacy* $\phi_v$ of a node $v \in V$ is the probability that $v$ is active, that is,

$$\phi_v = \Pr(X_{\mathrm{st}}(v) = 1) = \Pr(X_{sv} = 1)\Pr(X_{vt} = 1). \qquad (2.1)$$

In the interpretation of intermediacy, we focus on the ranking of nodes relative to each other. We do not consider the absolute values of intermediacy. For instance, suppose the intermediacy of node $v \in V$ is twice as high as the intermediacy of node $u \in V$. We then consider node $v$ to be more important than node $u$ in connecting the source $s$ and the target $t$. However, we do not consider node $v$ to be twice as important as node $u$.

We now present an analysis of the mathematical properties of intermediacy. The proofs of the mathematical results provided below can be found in appendix A.

## 2.1. Limit behaviour

To get a better understanding of intermediacy, we study the behaviour of intermediacy in two limit cases, namely the case in which the probability $p$ that an edge is active goes to 0 and the case in which the probability $p$ goes to 1. In each of the two cases, the ranking of the nodes in a graph based on intermediacy turns out to have a natural interpretation. The difference between the two cases is illustrated in figure 1a.

Let $\ell_v$ denote the length of the shortest source-target path going through node $v \in V$. The following theorem states that in the limit as the probability $p$ that an edge is active tends to 0, the ranking of nodes based on intermediacy coincides with the ranking based on $\ell_v$. Nodes located on shorter source-target paths are more intermediate than nodes located on longer source-target paths.

**Theorem 2.4.** *In the limit as the probability $p$ tends to 0, $\ell_u < \ell_v$ implies $\phi_u > \phi_v$.*

The intuition underlying this theorem is as follows. When the probability that an edge is active is close to 0, almost all edges are inactive. Consequently, almost all source-target paths are inactive as well. However, from a relative point of view, longer source-target paths are more likely to be inactive than shorter source-target paths. This means that nodes located on shorter source-target paths are more likely to be active than nodes located on longer source-target paths (even though for all nodes the probability of being active is close to 0). Nodes located on shorter source-target paths, therefore, have a higher intermediacy than nodes located on longer source-target paths.

We now consider the limit case in which the probability $p$ that an edge is active goes to 1. Let $\sigma_v$ denote the number of edge-independent source-target paths going through node $v \in V$. Theorem 2.5 states that in the limit as $p$ tends to 1, the ranking of nodes based on intermediacy coincides with the ranking based on $\sigma_v$. The larger the number of edge-independent source-target paths going through a node, the higher the intermediacy of the node.

**Theorem 2.5.** *In the limit as the probability p tends to 1, $\sigma_u > \sigma_v$ implies $\phi_u > \phi_v$.*

Intuitively, this theorem can be understood as follows. When the probability that an edge is active is close to 1, almost all edges are active. Consequently, almost all source-target paths are active as well, and so are almost all nodes. A node is inactive only if all source-target paths going through the node are inactive. If there are $\sigma$ edge-independent source-target paths that go through a node, this means that the node can be inactive only if there are at least $\sigma$ inactive edges. Consider two nodes $u$, $v \in V$. Suppose that the number of edge-independent source-target paths going through node $v$ is larger than the number of edge-independent source-target paths going through node $u$. In order to be inactive, node $v$ then requires more inactive edges than node $u$. This means that node $v$ is less likely to be inactive than node $u$ (even though for both nodes, the probability of being inactive is close to 0). Hence, node $v$ has a higher intermediacy than node $u$. More generally, nodes located on a larger number of edge-independent source-target paths have a higher intermediacy than nodes located on a smaller number of edge-independent source-target paths.

## 2.2. Parameter choice

The probability $p$ that an edge is active is a free parameter of intermediacy for which one needs to choose an appropriate value. The results presented above are concerned with the behaviour of intermediacy in the limit cases in which the probability $p$ tends to either 0 or 1. Figure 1$b$ provides some insight into the behaviour of intermediacy for values of the probability $p$ that are in between these two extremes. The figure shows two graphs. In the left graph, there is a direct path (i.e. a path of length 1) from node $u$ to node $v$. There are no indirect paths. In this graph, the probability that there is an active path from $u$ to node $v$ equals $p$. In the right graph, there is no direct path from node $u$ to node $v$, but there are $k$ indirect paths of length 2. Each of these paths has a probability of $p^2$ of being active. Consequently, the probability that there is at least one active path from node $u$ to node $v$ equals $1 - (1 - p^2)^k$. The bar chart in figure 1$b$ shows for different values of $k$ the values of $p$ for which the probability that there is an active path from node $u$ to node $v$ is higher (in orange) or lower (in grey) in the left graph than in the right graph. For instance, suppose that $k = 5$. For $p < 0.22$, the probability that there is an active path from node $u$ to node $v$ is higher in the left graph than in the right graph. For $p > 0.22$, the situation is the other way around. If the probability $p$ that an edge is active is set to 0.22, a direct path between two nodes is considered equally strong as five indirect paths of length 2. Based on figure 1$b$, one can set the probability $p$ to a value that one considers appropriate for a particular analysis.

## 2.3. Path addition and contraction

Next, we study two additional properties of intermediacy, the property of path addition and the property of path contraction. We show that both adding paths and contracting paths lead to an increase in intermediacy. Path addition and path contraction are important properties because they reflect the basic intuition underlying the idea of intermediacy. (Of course, in practice, paths cannot simply be added or contracted in a citation network. However, we can have two regions in a citation network that are topologically identical except for a path addition or a path contraction. Our theoretical analysis can be interpreted as an analysis comparing the intermediacy of the nodes in the two regions of the citation network.)

We start by considering the property of path addition. We define path addition as follows.

**Definition 2.6.** Consider a directed acyclic graph $G = (V, E)$ and two nodes $u$, $v \in V$ such that there does not exist a path from node $v$ to node $u$. *Path addition* is the operation in which a new path from node $u$ to node $v$ is added. Let $\ell$ denote the length of the new path. If $\ell = 1$, an edge $(u, v)$ is added. If $\ell > 1$, nodes $w_1, \ldots, w_{\ell-1}$ and edges $(u, w_1)$, $(w_1, w_2)$, $\ldots$, $(w_{\ell-2}, w_{\ell-1})$, $(w_{\ell-1}, v)$ are added.

This definition includes the condition that there does not exist a path from node $v$ to node $u$. This condition ensures that the graph $G$ will remain acyclic after adding a path. The following theorem states that adding a path increases intermediacy.

**Theorem 2.7.** *Consider a directed acyclic graph $G = (V, E)$, a source $s \in V$, and a target $t \in V$. In addition, consider two nodes $u$, $v \in V$ such that there does not exist a path from node $v$ to node $u$. Adding a path from node $u$ to node $v$ strictly increases the intermediacy $\phi_w$ of any node $w \in V$ located on a path from source $s$ to node $u$ or from node $v$ to target $t$.*

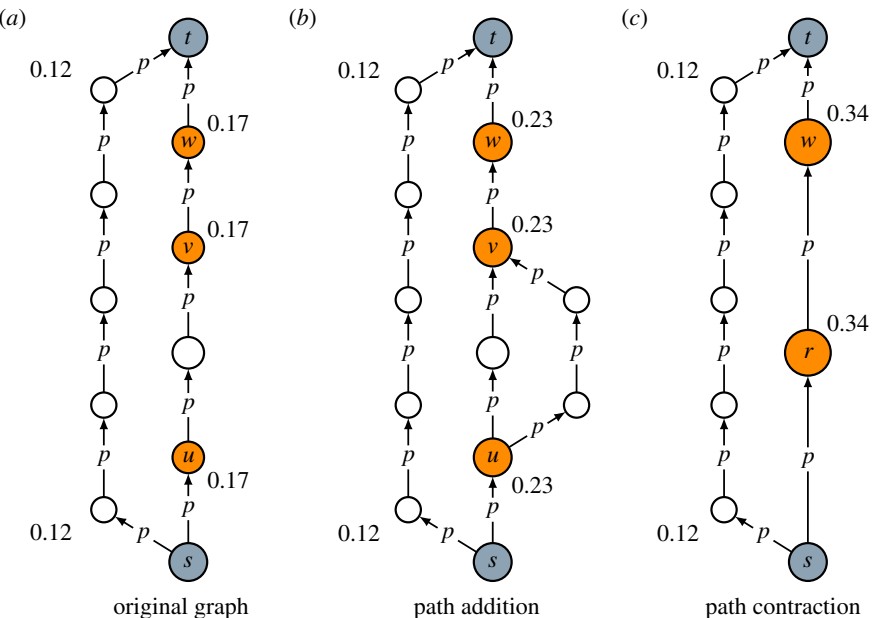

**Figure 2.** Illustration of the properties of path addition and path contraction. Comparing (*b*) to (*a*) shows how path addition increases intermediacy. Comparing (*c*) to (*b*) shows how path contraction increases intermediacy. For some nodes in (*a–c*), the intermediacy is reported, calculated using a value of 0.7 for the probability *p*.

Theorem 2.7 does not depend on the probability $p$. Adding a path always increases intermediacy, regardless of the value of $p$. To illustrate the theorem, consider figure 2*a,b*. The graph in figure 2*b* is identical to the one in figure 2*a* except that a path from node $u$ to node $v$ has been added. As can be seen, adding this path has increased the intermediacy of nodes located between source $s$ and node $u$ or between node $v$ and target $t$, including nodes $u$ and $v$ themselves. While the intermediacy of other nodes has not changed, the intermediacy of these nodes has increased from 0.17 to 0.23. This reflects the basic intuition that, after a path from node $u$ to node $v$ has been added, going from source $s$ to target $t$ through nodes $u$ and $v$ has become 'easier' than it was before. This means that nodes located between source $s$ and node $u$ or between node $v$ and target $t$ have become more important in connecting the source and the target. Consequently, the intermediacy of these nodes has increased.

We now consider the property of path contraction. We use $V_{uv}$ to denote the set of all nodes located on a path from node $u$ to node $v$, including nodes $u$ and $v$ themselves. Path contraction is then defined as follows.

**Definition 2.8.** Consider a directed acyclic graph $G = (V, E)$ and two nodes $u, v \in V$ such that there exists at least one path from node $u$ to node $v$. *Path contraction* is the operation in which all nodes in $V_{uv}$ are contracted. This means that the nodes in $V_{uv}$ are replaced by a new node $r$. Edges pointing from a node $w \notin V_{uv}$ to nodes in $V_{uv}$ are replaced by a single new edge $(w, r)$. Edges pointing from nodes in $V_{uv}$ to a node $w \notin V_{uv}$ are replaced by a single new edge $(r, w)$. Edges between nodes in $V_{uv}$ are removed.

The following theorem states that contracting paths increases intermediacy.

**Theorem 2.9.** *Consider a directed acyclic graph $G = (V, E)$, a source $s \in V$, and a target $t \in V$. In addition, consider two nodes $u, v \in V$ such that there exists at least one path from node $u$ to node $v$ and such that nodes in $V_{uv}$ do not have neighbours outside $V_{uv}$ except for incoming neighbours of node $u$ and outgoing neighbours of node $v$. Contracting paths from node $u$ to node $v$ strictly increases the intermediacy $\phi_w$ of any node $w \in V$ located on a path from source $s$ to node $u$ or from node $v$ to target $t$.*

Like theorem 2.7, theorem 2.9 does not depend on the probability $p$. Theorem 2.9 is illustrated in figure 2*b,c*. The graph in figure 2*c* is identical to the one in figure 2*b* except that paths from node $u$ to node $v$ have been contracted. As a result, there has been an increase in the intermediacy of nodes located between source $s$ and node $u$ or between node $v$ and target $t$, including nodes $u$ and $v$ themselves (which have been contracted into a new node $r$). While the intermediacy of other nodes has not changed, the intermediacy of these nodes has increased from 0.23 to 0.34. This reflects the basic intuition that, after paths from node $u$ to node $v$ have been contracted, going from source $s$ to

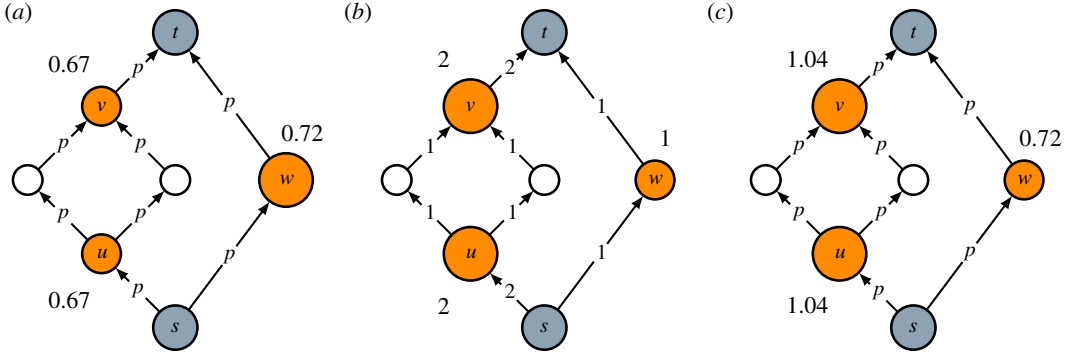

**Figure 3.** Comparison of (*a*) intermediacy, (*b*) main path analysis and (*c*) expected path count. For nodes *u*, *v* and *w*, the intermediacy (*a*), path count (*b*) and expected path count (*c*) are reported, using a value of 0.85 for the probability *p* in the calculation of intermediacy and expected path count.

target *t* through nodes *u* and *v* has become 'easier' than it was before. In other words, nodes located on a path from source *s* to target *t* going through nodes *u* and *v* have become more important in connecting the source and the target, and hence the intermediacy of these nodes has increased.

## 2.4. Alternative approaches

How does intermediacy differ from alternative approaches? We consider three alternative approaches. One is main path analysis [9]. This is the most commonly used approach for tracing the historical development of scientific knowledge in citation networks. The second alternative approach is the expected path count approach. Like intermediacy, the expected path count approach distinguishes between active and inactive edges and focuses on active source-target paths. While intermediacy considers the probability that there is at least one active source-target path going through a node, the expected path count approach considers the expected number of active source-target paths that go through a node. The third alternative approach is resistance [23–25]. Resistance is a measure of the distance between nodes in a graph. We use it to define an alternative to intermediacy.

Consider the graph shown in figure 3*a*. To get from source *s* to target *t*, one could take either a path going through nodes *u* and *v* or the path going through node *w*. Based on intermediacy, the latter path represents a stronger connection between the source and the target than the former one. This follows from the path contraction property.

Interestingly, main path analysis gives the opposite result, as can be seen in figure 3*b*. For each edge, the figure shows the search path count, which is the number of source-target paths that go through the edge. There are two source-target paths that go through (*s*, *u*) and (*v*, *t*), while all other edges are included only in a single source-target path. Because the search path counts of (*s*, *u*) and (*v*, *t*) are higher than the search path counts of (*s*, *w*) and (*w*, *t*), main path analysis favours paths going through nodes *u* and *v* over the path going through node *w*. This is exactly opposite to the result obtained using intermediacy. Figure 3*b* makes clear that main path analysis yields outcomes that violate the path contraction property. Main path analysis tends to favour longer paths over shorter ones. For the purpose of identifying publications that play an important role in connecting an older and a more recent publication, we consider this behaviour to be undesirable. There are various variants of main path analysis, which all show the same type of undesirable behaviour.

Instead of focusing on the probability of the existence of at least one active source-target path, as is done by intermediacy, one could also focus on the expected number of active source-target paths going through a node. This alternative approach, which we refer to as the expected path count approach, is illustrated in figure 3*c*. As can be seen in the figure, nodes *u* and *v* have a higher expected path count than node *w*. Paths going through nodes *u* and *v* may, therefore, be favoured over the path going through node *w*. Figure 3*c* shows that, unlike intermediacy, the expected path count approach does not have the path contraction property. Depending on the probability *p*, contracting paths may cause expected path counts to decrease rather than increase. Because the expected path count approach does not have the path contraction property, we do not consider this approach to be a suitable alternative to intermediacy.

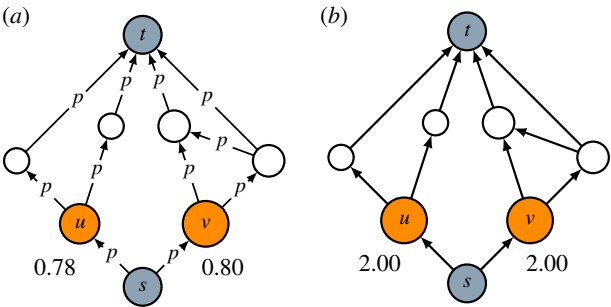

**Figure 4.** Comparison of (*a*) intermediacy and (*b*) resistance. For nodes *u* and *v*, the intermediacy (*a*) and resistance (*b*) are reported, using a value of 0.85 for the probability *p* in the calculation of intermediacy.

Finally, in figure 4, we illustrate the difference between intermediacy and resistance [23–25]. To get from source *s* to target *t*, one could take either a path going through node *u* or a path going through node *v*. Based on intermediacy, node *v* offers a stronger connection between the source and the target than node *u* (figure 4*a*). This follows from the path addition property. On the other hand, based on resistance, nodes *u* and *v* offer equally strong connections between the source and the target (figure 4*b*). Resistance is a measure of the distance between two nodes in a graph. Our interest focuses on the resistance between the source and the target. We define the resistance of a specific node as the resistance between the source and the target when only paths going through the node of interest are taken into account. Nodes *u* and *v* both have the same resistance of 2. According to the path addition property, node *v* should have a lower resistance than node *u*. (A lower resistance corresponds to a higher connectedness of the source and the target.) The equal resistance of nodes *u* and *v* shows that resistance does not have the path addition property.

# 3. Empirical analysis

We now present two case studies that serve as empirical illustrations of the use of intermediacy. Case 1 deals with the topic of community detection and its relationship with scientometric research. This case was selected because we are well acquainted with the topic and because we expect many readers of the present paper to be familiar with the topic as well. Case 2 deals with the topic of peer review. This case is of interest because it was examined using main path analysis in a recent paper by Batagelj *et al.* [26]. We consider this paper to be representative of the state of the art in main path analysis. Case 2, therefore, is well suited for demonstrating the differences between intermediacy and main path analysis.

In both case studies, the intermediacy of publications was calculated using the Monte Carlo algorithm presented in appendix B.

## 3.1. Case 1: community detection and scientometrics

We analyse how a method for community detection in networks ended up being used in the field of scientometrics to construct classification systems of scientific publications. In particular, we are interested in the historical development from Newman & Girvan [27] to Klavans & Boyack [28]. These are our target and source publications. Newman & Girvan [27] introduced a new measure for community detection in networks, known as modularity, while Klavans & Boyack [28] compared different ways in which modularity-based approaches can be used to identify communities in citation networks.

Our analysis relies on data from the Scopus database produced by Elsevier. We also considered the Web of Science database produced by Clarivate Analytics. However, many citation links relevant for our analysis are missing in Web of Science. There are also missing citation links in Scopus, but for Scopus the problem is less significant than for Web of Science. We refer to van Eck & Waltman [29] for a further discussion of the problem of missing citation links.

In the Scopus database, we found $n = 64\,223$ publications that are located on a citation path between our source and target publications. In total, we identified $m = 280\,033$ citation links between these

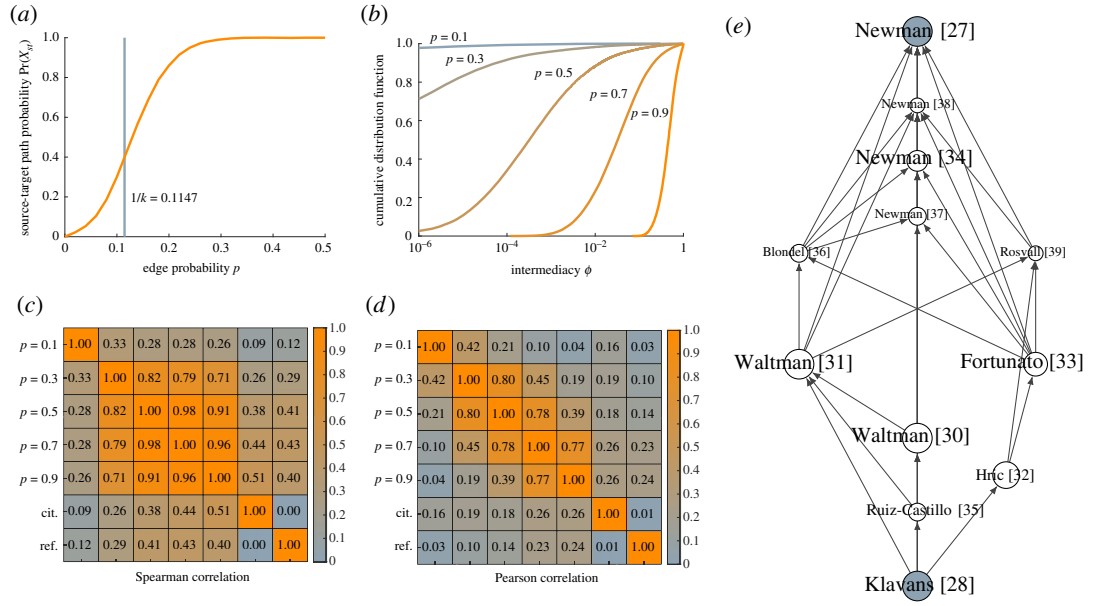

**Figure 5.** Results for case 1. (*a*) Probability of the existence of an active source-target path as a function of the parameter *p* and (*b*) cumulative distribution of intermediacy scores for different values of *p*. Spearman (*c*) and Pearson (*d*) correlations between intermediacy scores for different values of *p*, citation counts and reference counts. (*e*) Citation network of the top 10 most intermediate publications for $p = 0.1$. (Only the name of the first author is shown.)

publications. This means that on average each publication has $k = 2m/n \approx 8.72$ citation links, counting both incoming and outgoing links.

Figure 5*a* shows how the probability of the existence of an active path between the source and target publications depends on the parameter *p*. This probability increases from zero for $p = 0$ to almost one starting from $p = 0.25$. The vertical line indicates the value $p = 1/k$. At this value, traditional percolation theory for random graphs suggests that the probability that the source and target publications are connected becomes non-negligible [22]. When searching for a suitable value of *p*, the value $p = 1/k$ suggested by percolation theory may serve as a reasonable starting point. In our case, this yields $p \approx 1/8.72 \approx 0.11$, resulting in a probability of about 0.40 for the existence of an active source-target path.

For five different values of the parameter *p*, figure 5*b* shows the cumulative distribution of the intermediacy scores of our $n = 64\,223$ publications. As is to be expected, when *p* is close to zero, intermediacy scores are extremely small. On the other hand, when *p* is getting close to one, intermediacy scores also approach one.

Figure 5*c,d* shows Spearman and Pearson correlations between the intermediacy scores obtained for five different values of the parameter *p*. We consider intermediacy scores to be most useful from an ordinal perspective. From this point of view, Spearman correlations are more relevant than Pearson correlations, but for completeness, we report both types of correlations. The Spearman correlations show that values of 0.3, 0.5, 0.7 and 0.9 for *p* all yield fairly similar rankings of publications in terms of intermediacy. However, the ranking obtained for $p = 0.1$ is substantially different. Pearson correlations tend to be lower than Spearman correlations. Hence, even when different values of *p* yield similar rankings of publications, there usually does not exist a clear linear relationship between the intermediacy scores.

Figure 5*c,d* also shows correlations of intermediacy scores with citation counts and reference counts. The term *citation count* refers to the number of incoming citation links of a publication, while the term *reference count* refers to the number of outgoing citation links of a publication. Only citation links located on a citation path between the source and target publications are counted. Regardless of the value of *p*, intermediacy scores are not very strongly correlated with citation counts or reference counts.

Based on our expert knowledge of the topic under study, we found that the most useful results were obtained by setting the parameter *p* equal to 0.1. Table 1 lists the 10 publications with the highest intermediacy for $p = 0.1$. For each publication, the intermediacy is reported for five different values of *p*. In addition, the table also reports each publication's citation count and reference count. Figure 5*e* shows the citation network of the 10 most intermediate publications for $p = 0.1$.

**Table 1.** Top 10 most intermediate publications in case 1 for $p = 0.1$.

| | | $p$ | | | | | cit. | ref. |
|---|---|---|---|---|---|---|---|---|
| | | 0.1 | 0.3 | 0.5 | 0.7 | 0.9 | | |
| $t$ | Newman & Girvan [27] | 0.301 | 0.992 | 1.000 | 1.000 | 1.000 | 468 | 0 |
| $s$ | Klavans & Boyack [28] | 0.301 | 0.992 | 1.000 | 1.000 | 1.000 | 0 | 24 |
| 1 | Waltman & van Eck [30] | 0.061 | 0.376 | 0.656 | 0.878 | 0.988 | 2 | 27 |
| 2 | Waltman & van Eck [31] | 0.060 | 0.695 | 0.964 | 0.999 | 1.000 | 15 | 22 |
| 3 | Hric et al. [32] | 0.052 | 0.300 | 0.499 | 0.700 | 0.900 | 1 | 29 |
| 4 | Fortunato [33] | 0.037 | 0.629 | 0.972 | 1.000 | 1.000 | 73 | 154 |
| 5 | Newman [34] | 0.035 | 0.736 | 0.979 | 1.000 | 1.000 | 221 | 8 |
| 6 | Ruiz-Castillo & Waltman [35] | 0.024 | 0.360 | 0.624 | 0.847 | 0.981 | 2 | 24 |
| 7 | Blondel et al. [36] | 0.022 | 0.836 | 0.998 | 1.000 | 1.000 | 78 | 21 |
| 8 | Newman [37] | 0.021 | 0.851 | 0.999 | 1.000 | 1.000 | 138 | 18 |
| 9 | Newman [38] | 0.020 | 0.296 | 0.501 | 0.700 | 0.900 | 246 | 1 |
| 10 | Rosvall & Bergstrom [39] | 0.020 | 0.803 | 0.994 | 1.000 | 1.000 | 70 | 10 |

Using our expert knowledge to interpret the results presented in table 1 and figure 5*e*, we are able to trace how a method for community detection ended up in the scientometric literature. The two publications with the highest intermediacy [30,31] played a key role in introducing modularity-based approaches in the scientometric community. Waltman & van Eck [31] proposed the use of modularity-based approaches for constructing classification systems of scientific publications, while Waltman & van Eck [30] introduced an algorithm for implementing these modularity-based approaches. This algorithm can be seen as an improvement of the so-called Louvain algorithm introduced by Blondel et al. [36], which is also among the 10 most intermediate publications. Most of the other publications in table 1 and figure 5*e* are classical publications on community detection in general and modularity in particular. The publications by Newman all deal with modularity-based community detection. Rosvall & Bergstrom [39] proposed an alternative approach to community detection. They applied their approach to a citation network of scientific journals, which explains the connection with the scientometric literature. Fortunato [33] is a review of the literature on community detection. The intermediacy of this publication is probably strongly influenced by its large number of references. Hric et al. [32] is a more recent publication on community detection. This publication focuses on the challenges of evaluating the results produced by community detection methods. This issue is very relevant in a scientometric context, and therefore the publication was cited by our source publication [28]. Finally, there is one more scientometric publication in table 1 and figure 5*e*. This publication [35] is one of the first studies presenting a scientometric application of classification systems of scientific publications constructed using a modularity-based approach. The publication was also cited by our source publication.

The citation counts reported in table 1 show that some publications, especially the more recent ones, have a high intermediacy even though they have been cited only a very limited number of times. This makes clear that a ranking of publications based on intermediacy is quite different from a citation-based ranking of publications. The publications in table 1 that have a high intermediacy and a small number of citations do have a substantial number of references.

Finally, we compare the results obtained using intermediacy to the results given by main path analysis. The latter results, obtained using the original version of main path analysis [9] and using a more recent variant [12], can be found in electronic supplementary material, figures S1 and S2. Intermediacy and main path analysis provide completely different results. As shown in figure 5*e*, intermediacy yields a number of short paths between Newman & Girvan [27] in the community detection literature and Klavans & Boyack [28] in the scientometric literature. These paths go through well-known publications. On the other hand, main path analysis yields an extremely long path, going through more than 50 publications, most of which are not particularly well known. Despite our expert

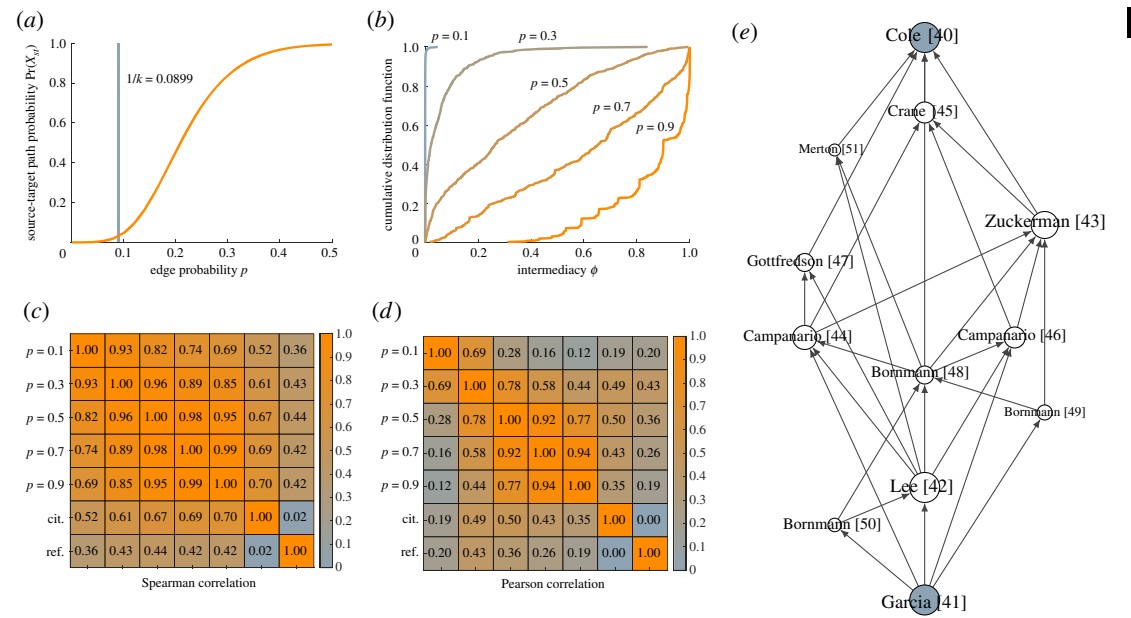

**Figure 6.** Results for case 2. (*a*) Probability of the existence of an active source-target path as a function of the parameter *p* and (*b*) cumulative distribution of intermediacy scores for different values of *p*. Spearman (*c*) and Pearson (*d*) correlations between intermediacy scores for different values of *p*, citation counts and reference counts. (*e*) Citation network of the top 10 most intermediate publications for *p* = 0.1. (Only the name of the first author is shown.)

understanding of both the community detection literature and the scientometric literature, there are many publications that we are not familiar with. Unlike the results obtained using intermediacy, we believe that the results given by main path analysis do not provide much insight into the historical development from Newman & Girvan [27] to Klavans & Boyack [28].

Case 2 presented next offers another comparison between intermediacy and main path analysis.

## 3.2. Case 2: peer review

In case 2, we analyse the literature on peer review. The analysis is based on data from the Web of Science database. We make use of the same data that was also used in a recent paper by Batagelj *et al.* [26].

We started with a citation network of 45 965 publications dealing with peer review. This is the citation network that was labelled CiteAcy by Batagelj *et al.* [26]. We selected Cole & Cole [40] and Garcia *et al.* [41] as our target and source publications. The main path analysis carried out by Batagelj *et al.* [26] suggests that these are central publications in the literature on peer review. For the purpose of our analysis, only publications located on a citation path between our source and target publications are of relevance. Other publications play no role in the analysis. We, therefore, restricted the analysis to the $n = 615$ publications located on a citation path from Garcia *et al.* [41] to Cole & Cole [40]. These publications are connected by $m = 3420$ citation links, resulting in an average of $k = 2m/n \approx 11.12$ citation links per publication.

As can be seen in figure 6*a*, percolation theory suggests a value of $1/k \approx 1/11.12 \approx 0.09$ for the parameter *p*. This is close to the value of 0.11 obtained in case 1. However, the probability of the existence of an active path between the source and target publications equals 0.03, which is much lower than the probability of 0.40 in case 1. Intermediacy scores tend to be higher in case 2 than in case 1. This can be seen by comparing figure 6*b* to figure 5*b*. We note that the former figure has a linear horizontal axis, while the horizontal axis in the latter figure is logarithmic. The Spearman and Pearson correlations are somewhat higher in case 2 (figure 6*c*,*d*) than in case 1 (figure 5*c*,*d*).

Table 2 lists the 10 publications with the highest intermediacy, where we use a value of 0.1 for the parameter *p*, like in table 1. Figure 6*e* shows the citation network of the 10 most intermediate publications. There are numerous paths in this citation network going from our source publication [41] to our target publication [40]. We regard these paths as the core paths between the source and target publications.

**Table 2.** Top 10 most intermediate publications in case 2 for $p = 0.1$.

| | | $p$ | | | | | cit. | ref. |
|---|---|---|---|---|---|---|---|---|
| | | 0.1 | 0.3 | 0.5 | 0.7 | 0.9 | | |
| $t$ | Cole & Cole [40] | 0.048 | 0.841 | 0.995 | 1.000 | 1.000 | 14 | 0 |
| $s$ | Garcia *et al.* [41] | 0.048 | 0.841 | 0.995 | 1.000 | 1.000 | 0 | 8 |
| 1 | Lee *et al.* [42] | 0.018 | 0.510 | 0.865 | 0.986 | 1.000 | 5 | 71 |
| 2 | Zuckerman & Merton [43] | 0.016 | 0.336 | 0.622 | 0.847 | 0.981 | 73 | 2 |
| 3 | Campanario [44] | 0.013 | 0.592 | 0.967 | 0.999 | 1.000 | 23 | 35 |
| 4 | Crane [45] | 0.009 | 0.270 | 0.498 | 0.700 | 0.900 | 34 | 1 |
| 5 | Campanario [46] | 0.009 | 0.517 | 0.952 | 0.999 | 1.000 | 15 | 30 |
| 6 | Gottfredson [47] | 0.008 | 0.320 | 0.622 | 0.847 | 0.981 | 26 | 2 |
| 7 | Bornmann [48] | 0.008 | 0.333 | 0.776 | 0.975 | 1.000 | 6 | 71 |
| 8 | Bornmann [49] | 0.007 | 0.259 | 0.500 | 0.700 | 0.900 | 1 | 20 |
| 9 | Bornmann [50] | 0.007 | 0.275 | 0.500 | 0.700 | 0.900 | 1 | 17 |
| 10 | Merton [51] | 0.005 | 0.243 | 0.497 | 0.701 | 0.901 | 29 | 1 |

The core paths shown in figure 6*e* can be compared to the results obtained by Batagelj *et al.* [26] using main path analysis. Different variants of main path analysis were used by Batagelj *et al.* [26]. Both using the original version of main path analysis [9] and using a more recent variant [12], the paths that were identified are rather lengthy, as can be seen in figs 9 and 10 in Batagelj *et al.* [26]. The shortest main paths include about 20 publications.

The above findings, together with the observations made in case 1, confirm the fundamental difference between intermediacy and main path analysis. Main path analysis tends to favour longer paths over shorter ones, whereas intermediacy has the opposite tendency.

Using the results presented in table 2 and figure 6*e*, experts on the topic of peer review could discuss the historical development of the literature on this topic. Since our own expertise on the topic of peer review is limited, we refrain from providing an interpretation of the results.

# 4. Conclusion

Citation networks provide valuable information for tracing the historical development of scientific knowledge. For this purpose, citation networks are usually analysed using main path analysis [9]. However, the idea of a main path is not very well understood. The algorithmic definition of a main path is clear, but the underlying conceptual motivation remains somewhat obscure. As we have shown in this paper, main path analysis has the tendency to favour longer paths over shorter ones. We regard this as a counterintuitive property that lacks a convincing justification.

Intermediacy, introduced in this paper, offers an alternative to main path analysis. It provides a principled approach for identifying publications that appear to play a major role in the historical development from an older to a more recent publication. The older publication and the more recent one are referred to as the target and the source, respectively. Publications with a high intermediacy are important in connecting the source and the target publication in a citation network. As we have shown, intermediacy has two intuitively desirable properties, referred to as path addition and path contraction. Because of the path contraction property, intermediacy tends to favour shorter paths over longer ones. This is a fundamental difference with main path analysis. Intermediacy also has a free parameter that can be used to fine-tune its behaviour. This parameter enables interpolation between two extremes. In one extreme, intermediacy identifies publications located on a shortest path between the source and the target publication. In the other extreme, it identifies publications located on the largest number of edge-independent source-target paths.

We have also examined intermediacy in two case studies. In the first case study, intermediacy was used to trace historical developments at the interface between the community detection literature and

the scientometric literature. This case study has shown that intermediacy yields results that make sense from our viewpoint as domain experts. In the second case study, intermediacy was applied to the literature on peer review. Both cases studies have demonstrated the strong preference of main path analysis for long paths.

There are various directions for further research. First of all, a more extensive mathematical analysis of intermediacy can be carried out, possibly resulting in an axiomatic foundation for intermediacy. Intermediacy can also be generalized to weighted graphs. In a citation network, a citation link may, for instance, be weighed inversely proportional to the total number of incoming or outgoing citation links of a publication. Another way to generalize intermediacy is to allow for multiple sources and targets. The ideas underlying intermediacy can also be used to develop other types of indicators for graphs, such as an indicator of the connectedness of two nodes in a graph. In empirical analyses, intermediacy can be applied not only in citation networks of scientific publications but for instance also in patent citation networks or in completely different types of networks, such as human mobility and migration networks, world trade networks, transportation networks, and passing networks in sports. Also, more comprehensive comparisons between intermediacy and main path analysis can be performed. The results of the two approaches can be evaluated in a systematic way based on input from domain experts.

Data accessibility. The data used in the first case study have been obtained from the Scopus database produced by Elsevier. Due to licence restrictions, the data cannot be made openly available. Readers can contact Elsevier to obtain the data (https://www.elsevier.com/solutions/scopus). The data used in the second case study have been obtained from the Web of Science database produced by Clarivate Analytics. Due to licence restrictions, the data cannot be made openly available. Readers can contact Clarivate Analytics to obtain the data (https://clarivate. com/products/web-of-science). The code used for computing the intermediacy is freely available online (https:// github.com/lovre/intermediacy).

Authors' contributions. L.Š., L.W., V.T. and N.J.E. designed research, L.Š., L.W., V.T. and N.J.E. performed research, L.Š., V.T. and N.J.E. analysed data and L.W. wrote the paper. All authors gave final approval for publication.

Competing interests. The authors have no conflicting interests to declare.

Funding. This work has been supported in part by the Slovenian Research Agency under the programmes P2-0359 and P5-0168, and by the European Union COST Action number CA15109.

Acknowledgements. The authors thank Vladimir Batagelj for sharing the data used to study the literature on peer review.

# Appendix A. Proofs

Below we provide the proofs of the theorems presented in the main text. We first need to introduce some additional notation. We use $\Pr(X_{uv})$ as a shorthand for $\Pr(X_{uv} = 1)$. To make explicit that this probability depends on a graph $G$, we write $\Pr(X_{uv} \mid G)$. Furthermore, we use $A_e$ to indicate whether an edge $e$ is active. Hence, $A_e = 1$ if edge $e$ is active and $A_e = 0$ if edge $e$ is not active.

## A.1. Limit behaviour

*Proof of theorem 2.4.* Let $m = |E|$ denote the number of edges in the graph $G$. Suppose that the $m$ edges are split into two sets, one set of $M$ edges and another set of $m - M$ edges. The probability that the edges in the former set are all active while the edges in the latter set are all inactive equals

$$P_M = p^M (1 - p)^{m-M}.$$

Consider a node $v \in V$. The shortest source-target path that goes through node $v$ has a length of $\ell_v$. This means that at least $\ell_v$ edges need to be active in order to obtain an active source-target path that goes through node $v$. Hence, the probability that there is an active source-target path that goes through node $v$ can be written as

$$\phi_v = \sum_{i=\ell_v}^{m} n_{vi} P_i,$$

where $n_{vi} > 0$ for all $i = \ell_v, \ldots, m$. Note that this probability equals the intermediacy of node $v$. Now consider two nodes $u, v \in V$ with $\ell_u < \ell_v$. In the limit as $p$ tends to 0, $\phi_u$ and $\phi_v$ both tend to 0.

However, they do so at different rates. More specifically, in the limit as $p$ tends to 0, we have

$$\lim_{p\to 0}\frac{\phi_v}{\phi_u} = \lim_{p\to 0}\frac{\sum_{i=\ell_v}^{m} n_{vi}P_i}{\sum_{i=\ell_u}^{m} n_{ui}P_i}$$

$$= \lim_{p\to 0}\frac{\sum_{i=\ell_v}^{m} n_{vi}P_i/P_{\ell_u}}{\sum_{i=\ell_u}^{m} n_{ui}P_i/P_{\ell_u}}$$

$$= \lim_{p\to 0}\frac{\sum_{i=\ell_v}^{m} n_{vi}p^{i-\ell_u}(1-p)^{\ell_u-i}}{\sum_{i=\ell_u}^{m} n_{ui}p^{i-\ell_u}(1-p)^{\ell_u-i}}$$

$$= \frac{0}{n_{u\ell_u}}$$

$$= 0.$$

Hence, in the limit as $p$ tends to 0, $\phi_u > \phi_v$. ∎

*Proof of theorem 2.5.* Let $m = |E|$ denote the number of edges in the graph $G$, and let $q$ denote the probability that an edge is inactive, that is, $q = 1 - p$. Suppose that the $m$ edges are split into two sets, one set of $M$ edges and another set of $m - M$ edges. The probability that the edges in the former set are all inactive while the edges in the latter set are all active equals

$$Q_M = q^M(1-q)^{m-M}.$$

Consider a node $v \in V$. There are $\sigma_v$ edge-independent source-target paths that go through node $v$. This means that at least $\sigma_v$ edges need to be inactive in order for there to be no active source-target path that goes through node $v$. Hence, the probability that there is no active source-target path that goes through node $v$ can be written as

$$\Phi_v = \sum_{i=\sigma_v}^{m} n_{vi}Q_i,$$

where $n_{vi} > 0$ for all $i = \sigma_v, , m$. Note that the intermediacy of node $v$ equals 1 minus this probability, that is, $\phi_v = 1 - \Phi_v$. Now consider two nodes $u, v \in V$ with $\sigma_u > \sigma_v$. In the limit as $p$ tends to 1, $\Phi_u$ and $\Phi_v$ both tend to 0. However, they do so at different rates. More specifically, in the limit as $p$ tends to 1, we have

$$\lim_{p\to 1}\frac{\Phi_u}{\Phi_v} = \lim_{p\to 1}\frac{\sum_{i=\sigma_u}^{m} n_{ui}Q_i}{\sum_{i=\sigma_v}^{m} n_{vi}Q_i}$$

$$= \lim_{p\to 1}\frac{\sum_{i=\sigma_u}^{m} n_{ui}Q_i/Q_{\sigma_v}}{\sum_{i=\sigma_v}^{m} n_{vi}Q_i/Q_{\sigma_v}}$$

$$= \lim_{p\to 1}\frac{\sum_{i=\sigma_u}^{m} n_{ui}q^{i-\sigma_v}(1-q)^{\sigma_v-i}}{\sum_{i=\sigma_v}^{m} n_{vi}q^{i-\sigma_v}(1-q)^{\sigma_v-i}}$$

$$= \frac{0}{n_{v\sigma_v}}$$

$$= 0.$$

Hence, in the limit as $p$ tends to 1, $\Phi_u < \Phi_v$, which implies that $\phi_u > \phi_v$. ∎

## A.2. Path addition and path contraction

*Proof of theorem 2.7.* Suppose that node $w$ is located on a path from source $s$ to node $u$. Let $H$ denote the graph obtained after the path from node $u$ to node $v$ has been added, and let $E_{uv}$ denote the set of newly added edges. The intermediacy of node $w$ in graph $G$ can be factorized as $\phi_w(G) = \Pr(X_{sw} \mid G)\Pr(X_{wt} \mid G)$. Similarly, for graph $H$, we have $\phi_w(H) = \Pr(X_{sw} \mid H)\Pr(X_{wt} \mid H)$. Clearly, $\Pr(X_{sw} \mid G) = \Pr(X_{sw} \mid H)$, since the paths from node $s$ to node $w$ are identical in graphs $G$ and $H$. Furthermore, $\Pr(X_{wt} \mid G) = \Pr(X_{wt} \mid H \text{ and } \forall e \in E_{uv} : A_e = 0)$. Since $\Pr(X_{wt} \mid H \text{ and } \forall e \in E_{uv} : A_e = 0) < \Pr(X_{wt} \mid H)$, it follows that $\Pr(X_{wt} \mid G) < \Pr(X_{wt} \mid H)$. This means that $\phi_w(G) < \phi_w(H)$.

An analogous proof can be given if node $w$ is located on a path from node $v$ to target $t$. ∎

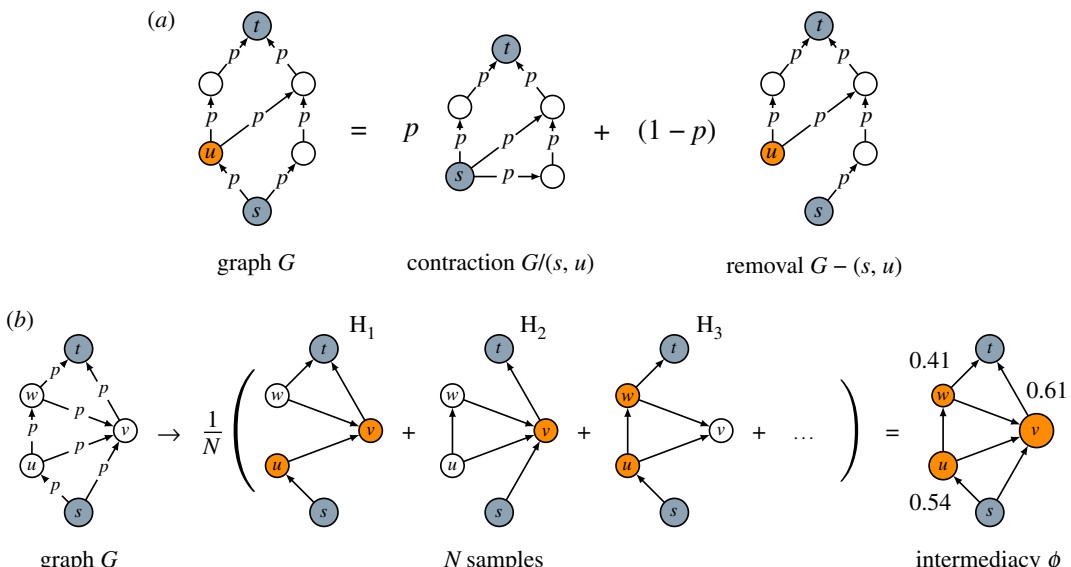

**Figure 7.** Illustration of the calculation of intermediacy using the exact algorithm (*a*) and using the Monte Carlo algorithm for $p = 0.7$ (*b*).

*Proof of theorem 2.9.* Suppose that node $w$ is located on a path from source $s$ to node $u$. Let $H$ denote the graph obtained after paths from node $u$ to node $v$ have been contracted, and let $E_{uv}$ denote the set of all edges between nodes in $V_{uv}$. The intermediacy of node $w$ in graph $G$ can be factorized as $\phi_w(G) = \Pr(X_{sw} \mid G)\Pr(X_{wt} \mid G)$. Similarly, for graph $H$, we have $\phi_w(H) = \Pr(X_{sw} \mid H)\Pr(X_{wt} \mid H)$. Clearly, $\Pr(X_{sw} \mid G) = \Pr(X_{sw} \mid H)$, since the paths from node $s$ to node $w$ are identical in graphs $G$ and $H$. Furthermore, because nodes in $V_{uv}$, except for nodes $u$ and $v$, do not have neighbours outside $V_{uv}$, we have $\Pr(X_{wt} \mid H) = \Pr(X_{wt} \mid G \text{ and } \forall e \in E_{uv} : A_e = 1)$. Since $\Pr(X_{wt} \mid G \text{ and } \forall e \in E_{uv} : A_e = 1) > \Pr(X_{wt} \mid G)$, it follows that $\Pr(X_{wt} \mid H) > \Pr(X_{wt} \mid G)$. This means that $\phi_w(H) > \phi_w(G)$.

An analogous proof can be given if node $w$ is located on a path from node $v$ to target $t$. ∎

# Appendix B. Algorithms

Intermediacy depends on the probability that there exists a path between two nodes in a graph. Determining this probability is known as the problem of network reliability. This problem is NP-hard [52]. Below we provide an outline of an exact algorithm for calculating intermediacy. Because of its exponential run-time, the exact algorithm can be used only in relatively small graphs. We, therefore, also propose a Monte Carlo algorithm that approximates intermediacy.

## B.1. Exact algorithm

The exact algorithm, illustrated in figure 7*a*, is based on contraction and deletion of edges [53]. Suppose we have a graph $G = (V, E)$. The probability that there exists a path between two nodes $u, v \in V$ can be written as

$$\Pr(X_{uv} \mid G) = p\Pr(X_{uv} \mid G/e) + (1 - p)\Pr(X_{uv} \mid G - e), \tag{B 1}$$

where $G/e$ denotes the contraction of an edge $e \in E$ and $G - e$ denotes the deletion of an edge $e \in E$. Edge contraction must respect reachability [54]. Equation (B 1) yields a recursive algorithm for calculating $\Pr(X_{uv})$. For a node $v \in V$, this algorithm can be used to calculate $\Pr(X_{sv})$ and $\Pr(X_{vt})$. The intermediacy $\phi_v$ of node $v$ is then given by equation (2.1). We are usually interested in calculating the intermediacy of all nodes in a graph $G$, not just of one specific node. This can be performed efficiently by calculating $\Pr(X_{sv})$ and $\Pr(X_{vt})$ for all nodes $v \in V$ in a single recursion.

The run-time of the exact algorithm is exponential in the number of edges $m$. The algorithm has a complexity of $\mathcal{O}(2^m)$. In the special case of a so-called series–parallel graph, the run-time of the algorithm can be reduced from exponential to polynomial [55].

## B.2. Monte Carlo algorithm

The Monte Carlo algorithm, illustrated in figure 7$b$, is quite straightforward. Suppose we have a graph $G = (V, E)$ and we are interested in the intermediacy $\phi_v$ of a node $v \in V$. A subgraph $H$ can be obtained by sampling the edges in the graph $G$, where each edge $e \in E$ is sampled with probability $p$. Given a subgraph $H$, it can be determined whether in this subgraph node $v$ is located on a path from source $s$ to target $t$. We sample $N$ subgraphs $H_1, \ldots, H_N$. We then approximate the intermediacy of node $v$ by $\phi_v \approx \frac{1}{N} \sum_{i=1}^{N} I_{st}(v \mid H_i)$, where $I_{st}(v \mid H_i)$ equals 1 if there exists a path from source $s$ to target $t$ going through node $v$ in graph $H_i$ and 0 otherwise.

The Monte Carlo algorithm can be implemented efficiently by simultaneously sampling subgraphs and checking path existence. To do so, we perform a probabilistic depth-first search. We maintain a stack of nodes that still need to be visited. We start by pushing source $s$ to the stack. We then keep popping nodes from the stack until the stack is empty. When a node $v$ has been popped from the stack, we determine for each of its outgoing edges whether the edge is active. An edge is active with probability $p$. If an edge $(v, u)$ is active and if node $u$ is not yet on the stack, then node $u$ is pushed to the stack. At some point, target $t$ may be reached, resulting in the identification of nodes that are located on a path from source $s$ to target $t$. This implementation of the Monte Carlo algorithm is especially fast for smaller values of the probability $p$. The run-time of the Monte Carlo algorithm is linear in the number of edges $m$.

In this paper, we use a Java implementation of the Monte Carlo algorithm. The source code is available at https://github.com/lovre/intermediacy [56].

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
