## [Reviewer comments · Royal Society Open Science]

Review History

RSOS-190207.R0 (Original submission)

Review form: Reviewer 1

Is the manuscript scientifically sound in its present form?

Yes

Are the interpretations and conclusions justified by the results?

Yes

Is the language acceptable?

Yes

Is it clear how to access all supporting data?

Yes

Do you have any ethical concerns with this paper?

No

Have you any concerns about statistical analyses in this paper?

No

Recommendation?

Accept with minor revision (please list in comments)

Comments to the Author(s)

The paper introduces the concept of intermediacy, a metric to identify major publications (knowledge) that have led from one paper to another. Overall, I enjoyed reading this work - the text was concise, and I appreciated the clear explanations and use of figures. I also think that intermediacy as a concept has value. There were a few locations where I think the text requires clarification, but I believe that is easy to fix. My main comments apply to most papers that present a new metric: it's important to clearly state the need for it, and how it complements existing metrics. Currently the paper briefly dismisses main path analysis claiming that the results are counter-intuitive (and therefore it is necessary to develop a new metric). However, the paper lacks a rigorous literature review and, IMHO, this is the wrong strategy. It would be better to provide a meaningful discussion of the metrics available, highlighting their strength and weaknesses when used in different forms of analysis. This should then naturally lead to a 'gap in the market' which intermediacy fills. Of course, you should then highlight the strength/weaknesses of intermediacy, explaining how future researchers should interpret the results if they use it. In the below text I have tried to provide more specific comments that could help in this goal. But congratulations on a nice piece of work, and I hope the feedback can be of use.

- I like the introduction - it is clear and concise. However, it would be good if you could add some further motivational discussion. You mention main path analysis, and then immediately present your approach. Yet, it is not clear why your approach is required, i.e. what are the limitations that you seek to address? A few sentences describing the problems with prior main path analysis techniques would be useful for the reader.

- "we will show that main path analysis tends to favor longer citation paths over shorter ones, whereas intermediacy has the opposite tendency" - again, it would be useful to add some further context here: what is the benefit of your technique favouring shorter paths? You quickly dismiss main path analysis as counter-intuitive, but it would be better if you could provide a more rigorous analysis. Otherwise, it is not clear to a reader why it is necessary to develop the intermediacy metric, and what analytical benefits it brings. Just to be clear, I *do* think there is a benefit; my suggestion is simply to make this explicit within the early parts of the paper. A positive step would be to add a Background section to describe (and critique) past metrics/studies.

- "we argue that intermediacy yields better results than main path analysis". To address the above comment, you could expand on this assertion. Why do you argue this - could you briefly summarise your evidence here? I would also mention that (hopefully) other scientometrics researchers will use intermediacy in their studies. Making a strong case for what extra value it brings to a bibliometric study would help in this.

- Minor: I would suggest making the definition of "active edge" more explicit. Definition 2.2 covers what an "active path" is, but this relies on the assumption that the reader understands an "active edge". After reading further on, this obviously becomes clear. But it would be useful to have a brief explicit definition before definition 2.2.

- When I started reading Section 2(B), I assumed that this would offer guidelines for setting p . However, the conclusion of "one can set the probability p to a value that one considers appropriate for a particular analysis" does not really answer the more meaningful question that

the section poses: how should I select an appropriate ρ for my particular analysis? You later use $1/k$ - is this your general recommendation?

- It is clear what the implications of changing ρ are, but setting it seems to require the researcher to know what path properties they are looking for in their citation graph (i.e. shortest path or num independent paths). Do you have any recommendations or insight here, because selecting a "wrong" ρ would obviously impact the outcome. IMHO, this query could be addressed in one of two ways. Either (1) You could provide a methodology or (approximate) guideline regarding how people could select ρ , or (2) You could give some clear insight into how to interpret results based on the value of ρ . The latter could be done in a table, which summarises implications for certain ranges of ρ . Again, if the answer is $1/k$, you could state this here.

- Minor: p6, line 28,48; p10, line 15,23,34; p11, line 33,35,38: table/fig need uppercase letters (you switch between upper and lower case)

- Page 7, "For the purpose of identifying publications that play an important role in connecting an older and a more recent publication, we consider this behavior to be undesirable". I found this statement a bit confusing. There are a few statements critiquing main path analysis, but the text tends to be brief and dismissive. Incidentally, framing things as being "undesirable" might be the wrong tactic - it would be better to highlight the different insights that can be gleaned from the two (complementary?) approaches. If you really think main path analysis is scientifically wrong, you should provide clear evidence for this.

- With the above in mind, it would be good to compare your results directly against main path analysis. The second case study uses [22], but it isn't mentioned in the first case study. You could then use this to expand your claims. You could explicitly state how the results differ, and the complementary (or superior?) nature of the insights gleaned. As I said before, I don't think you necessarily need to criticise main path analysis - instead, you could just more clearly highlight the value of using *both* techniques within a bibliometric study.

- I was curious why you limited yourselves to these two specific case studies with single pairs of paper? Could some details be added here? At first, I thought it was because you had the domain expertise to evaluate the correctness of the outcome. However, in Case Study 1 you just briefly discuss the papers, and in Case Study 2 you say that you don't have the expertise to comment on it. It would be valuable to actually try to quantify the correctness of your results - for instance, by performing a survey of relevant researchers to judge accuracy. I wouldn't expect this for this paper, but might be interesting future work.

- Again, without wishing to strawman the paper, why did you choose not to scale-out your empirical analysis? For instance, you could start with a domain (community detection) and then try many different $\langle s, d \rangle$ pairs, so that you can explore more generalisable trends. When you only present 2 specific case studies, it naturally raises questions related to how applicable this might be to other domains (or even other pairs of papers). Is this a fair comment? If not, it would be good to briefly explain why such an approach is unsuitable for your goals.

- I liked the short discussion about how intermediacy differs from citation rates in some cases. More generally, I think these discussions are valuable (in any of paper which introduces a new metric). I would recommend expanding this type of discussion, such that you better contrast intermediacy with other relevant metrics to clearly state why people working in scientometrics should use it in the future. For example, you could add a table that summarises the strength/weaknesses of several metrics (citation count etc) including intermediacy.

- Quick question: I wasn't clear why you felt intermediacy scores were more useful from an ordinal perspective? Indeed, Spearman often shows stronger correlations than Pearson. But is your ordinal statement simply driven by the the higher correlations you observe, or was there something more fundamental?

- Minor: Could you add the X labels for the heatmaps, it would just make it easier to read.

- Minor: p10,line 12: character error.

Review form: Reviewer 2 (Saeed-UI Hassan)

Is the manuscript scientifically sound in its present form?

No

Are the interpretations and conclusions justified by the results?

Yes

Is the language acceptable?

Yes

Is it clear how to access all supporting data?

Not Applicable

Do you have any ethical concerns with this paper?

No

Have you any concerns about statistical analyses in this paper?

No

Recommendation?

Major revision is needed (please make suggestions in comments)

Comments to the Author(s)

This is an interesting well-written paper that seeks to trace the historical development of scientific knowledge. The authors show that the proposed intermediacy measure outperforms main path analysis – a known approach for tracing historical developments in citation networks. I have the following major concerns:

- 1) Several measures exist in graph theory literature such as betweenness centrality, closeness etc. that are similar to the proposed intermediacy measure. I would like authors to justify why these known measures are not suited in this context? also, justify the proposal for this new measure.
- 2) The authors claim that the intermediacy increases if a given path is contracted or extend. However, the authors fail to give the practical implication of this phenomenon in a citation network, which is a fixed graph.
- 3) Another well-known measure that can measure the quality and strength of links is Kirchhoff index, I would ask authors to show that the intermediacy measure outperforms Kirchhoff index in capturing the historical development.

4) The manuscript contains trivial statements which are written as detailed theorems in the text, and this can be avoided by merely citing the relevant literature.

5) Attached (Appendix A) find a white paper that has been compiled for a relevant project, containing several classic approaches to do what authors are trying with intermediacy measure. I would ask the authors to justify mathematically and/or empirically how the proposed measure is comparable with existing indices.

Decision letter (RSOS-190207.R0)

21-Jun-2019

Dear Professor Šubelj

On behalf of the Editors, I am pleased to inform you that your Manuscript RSOS-190207 entitled "Intermediacy of publications" has been accepted for publication in Royal Society Open Science subject to minor revision in accordance with the referee suggestions. Please find the referees' comments at the end of this email.

The reviewers and handling editors have recommended publication, but also suggest some minor revisions to your manuscript. Therefore, I invite you to respond to the comments and revise your manuscript.

- Ethics statement

- Data accessibility

<http://datadryad.org/submit?journalID=RSOS&manu=RSOS-190207>

- Competing interests

- Authors' contributions

- Acknowledgements

- Funding statement

Because the schedule for publication is very tight, it is a condition of publication that you submit the revised version of your manuscript before 30-Jun-2019. Please note that the revision deadline will expire at 00.00am on this date. If you do not think you will be able to meet this date please let me know immediately.

Kind regards,
Lianne Parkhouse
Editorial Coordinator
Royal Society Open Science
openscience@royalsociety.org

on behalf of Dr Jon Crowcroft (Associate Editor) and Marta Kwiatkowska (Subject Editor)
openscience@royalsociety.org

Associate Editor Comments to Author (Dr Jon Crowcroft):

thank you for the submission - as you can see the reviewers have some suggestions for revisions - i think these are really minor, so please follow what they suggest, and well done!

Reviewer comments to Author:

Reviewer: 1:

The paper introduces the concept of intermediacy, a metric to identify major publications (knowledge) that have led from one paper to another. Overall, I enjoyed reading this work - the text was concise, and I appreciated the clear explanations and use of figures. I also think that intermediacy as a concept has value. There were a few locations where I think the text requires clarification, but I believe that is easy to fix. My main comments apply to most papers that present a new metric: it's important to clearly state the need for it, and how it complements existing metrics. Currently the paper briefly dismisses main path analysis claiming that the results are counter-intuitive (and therefore it is necessary to develop a new metric). However, the paper lacks a rigorous literature review and, IMHO, this is the wrong strategy. It would be better to provide a meaningful discussion of the metrics available, highlighting their strength and weaknesses when used in different forms of analysis. This should then naturally lead to a 'gap in the market' which intermediacy fills. Of course, you should then highlight the strength/weaknesses of intermediacy, explaining how future researchers should interpret the results if they use it. In the below text I have tried to provide more specific comments that could help in this goal. But congratulations on a nice piece of work, and I hope the feedback can be of use.

- I like the introduction - it is clear and concise. However, it would be good if you could add some further motivational discussion. You mention main path analysis, and then immediately present your approach. Yet, it is not clear why your approach is required, i.e. what are the limitations that you seek to address? A few sentences describing the problems with prior main path analysis techniques would be useful for the reader.

- "we will show that main path analysis tends to favor longer citation paths over shorter ones, whereas intermediacy has the opposite tendency" - again, it would be useful to add some further context here: what is the benefit of your technique favouring shorter paths? You quickly dismiss main path analysis as counter-intuitive, but it would be better if you could provide a more rigorous analysis. Otherwise, it is not clear to a reader why it is necessary to develop the intermediacy metric, and what analytical benefits it brings. Just to be clear, I *do* think there is a benefit; my suggestion is simply to make this explicit within the early parts of the paper. A positive step would be to add a Background section to describe (and critique) past metrics/studies.

- "we argue that intermediacy yields better results than main path analysis". To address the above comment, you could expand on this assertion. Why do you argue this - could you briefly summarise your evidence here? I would also mention that (hopefully) other scientometrics researchers will use intermediacy in their studies. Making a strong case for what extra value it brings to a bibliometric study would help in this.

- Minor: I would suggest making the definition of "active edge" more explicit. Definition 2.2 covers what an "active path" is, but this relies on the assumption that the reader understands an "active edge". After reading further on, this obviously becomes clear. But it would be useful to have a brief explicit definition before definition 2.2.

- When I started reading Section 2(B), I assumed that this would offer guidelines for setting p . However, the conclusion of "one can set the probability p to a value that one considers appropriate for a particular analysis" does not really answer the more meaningful question that the section poses: how should I select an appropriate p for my particular analysis? You later use $1/k$ - is this your general recommendation?

- It is clear what the implications of changing p are, but setting it seems to require the researcher to know what path properties they are looking for in their citation graph (i.e. shortest path or num independent paths). Do you have any recommendations or insight here, because selecting a "wrong" p would obviously impact the outcome. IMHO, this query could be addressed in one of two ways. Either (1) You could provide a methodology or (approximate) guideline regarding how people could select p , or (2) You could give some clear insight into how to interpret results based on the value of p . The latter could be done in a table, which summarises implications for certain ranges of p . Again, if the answer is $1/k$, you could state this here.

- Minor: p6, line 28,48; p10, line 15,23,34; p11, line 33,35,38: table/fig need uppercase letters (you switch between upper and lower case)

- Page 7, "For the purpose of identifying publications that play an important role in connecting an older and a more recent publication, we consider this behavior to be undesirable". I found this statement a bit confusing. There are a few statements critiquing main path analysis, but the text tends to be brief and dismissive. Incidentally, framing things as being "undesirable" might be the wrong tactic - it would be better to highlight the different insights that can be gleaned from the two (complementary?) approaches. If you really think main path analysis is scientifically wrong, you should provide clear evidence for this.

- With the above in mind, it would be good to compare your results directly against main path analysis. The second case study uses [22], but it isn't mentioned in the first case study. You could then use this to expand your claims. You could explicitly state how the results differ, and the complementary (or superior?) nature of the insights gleaned. As I said before, I don't think you necessarily need to criticise main path analysis - instead, you could just more clearly highlight the value of using *both* techniques within a bibliometric study.

- I was curious why you limited yourselves to these two specific case studies with single pairs of paper? Could some details be added here? At first, I thought it was because you had the domain expertise to evaluate the correctness of the outcome. However, in Case Study 1 you just briefly discuss the papers, and in Case Study 2 you say that you don't have the expertise to comment on it. It would be valuable to actually try to quantify the correctness of your results - for instance, by performing a survey of relevant researchers to judge accuracy. I wouldn't expect this for this paper, but might be interesting future work.

- Again, without wishing to strawman the paper, why did you choose not to scale-out your empirical analysis? For instance, you could start with a domain (community detection) and then try many different $\langle s, d \rangle$ pairs, so that you can explore more generalisable trends. When you only present 2 specific case studies, it naturally raises questions related to how applicable this might be to other domains (or even other pairs of papers). Is this a fair comment? If not, it would be good to briefly explain why such an approach is unsuitable for your goals.

- I liked the short discussion about how intermediacy differs from citation rates in some cases. More generally, I think these discussions are valuable (in any of paper which introduces a new metric). I would recommend expanding this type of discussion, such that you better contrast

intermediacy with other relevant metrics to clearly state why people working in scientometrics should use it in the future. For example, you could add a table that summarises the strength/weaknesses of several metrics (citation count etc) including intermediacy.

- Quick question: I wasn't clear why you felt intermediacy scores were more useful from an ordinal perspective? Indeed, Spearman often shows stronger correlations than Pearson. But is your ordinal statement simply driven by the the higher correlations you observe, or was there something more fundamental?

- Minor: Could you add the X labels for the heatmaps, it would just make it easier to read.

- Minor: p10,line 12: character error.

Reviewer: 2:

This is an interesting well-written paper that seeks to trace the historical development of scientific knowledge. The authors show that the proposed intermediacy measure outperforms main path analysis – a known approach for tracing historical developments in citation networks. I have the following major concerns:

- 1) Several measures exist in graph theory literature such as betweenness centrality, closeness etc. that are similar to the proposed intermediacy measure. I would like authors to justify why these known measures are not suited in this context? also, justify the proposal for this new measure.
- 2) The authors claim that the intermediacy increases if a given path is contracted or extend. However, the authors fail to give the practical implication of this phenomenon in a citation network, which is a fixed graph.
- 3) Another well-known measure that can measure the quality and strength of links is Kirchhoff index, I would ask authors to show that the intermediacy measure outperforms Kirchhoff index in capturing the historical development.
- 4) The manuscript contains trivial statements which are written as detailed theorems in the text, and this can be avoided by merely citing the relevant literature.
- 5) Attached find a white paper that has been compiled for a relevant project, containing several classic approaches to do what authors are trying with intermediacy measure. I would ask the authors to justify mathematically and/or empirically how the proposed measure is comparable with existing indices.

Author's Response to Decision Letter for (RSOS-190207.R0)

See Appendix B.

Decision letter (RSOS-190207.R1)

22-Nov-2019

Dear Professor Šubelj,

It is a pleasure to accept your manuscript entitled "Intermediacy of publications" in its current form for publication in Royal Society Open Science. The comments of the reviewer(s) who reviewed your manuscript are included at the foot of this letter.

Kind regards,

on behalf of Dr Jon Crowcroft (Associate Editor) and Marta Kwiatkowska (Subject Editor)
openscience@royalsociety.org

Appendix A

Defending Network Structure Using Trusted Components

Connectivity between nodes is a characteristic phenomenon of any networked system. It directly implies the structural robustness of network against node and link failures. As a result of some node or link failures, that is removal of such nodes or links, a network might become disconnected which might disrupt network operations causing loss and damage. Often times, the damage incurred is correlated to the number of components into which a network is divided and the sizes of the resulting components. To quantify the consequences of node or edge link removals, and consequently characterize the resilience of network structure against such removals, various measures have been identified in literature.

Various terms have been used (interchangeably) in reference to graph's ability to retain and preserve its structure as a result of node and edge removals. Some of them are as reliability, stability [1], vulnerability (e.g., [2, 3, 4], survivability, robustness of (graphs) network structure.

There are measures that aim to quantify both the effort required to cause the damage; such as the number of nodes or edges that need to be removed; as well as the extent of damage (change in network structure), such as the number or sizes of components into which the network is broken.

Based on the particular aspect or definition of change in structure, there are several measures.

Some of these parameters include *vertex (edge) connectivity* [5], *toughness* [6], *tenacity* [7], *integrity* [8], *binding number* [9], *fragmentability* [10], *average connectivity* [11] *isoperimetric number* [12], *expansion ratio* [13], *r-robustness* [14], and others [15].

1 Notations

We consider an undirected graph $G(V, E)$ with the node set V and the edge set E . Here, we consider node removals in undirected graphs, but similar notions can be conceived for the case of edge removals. Moreover, the graph obtained after removing a subset of nodes is referred to as the *residual graph*, denoted by $G'(V', E')$. The number of nodes in a graph is the *order* of the graph, whereas, the number of edges is the *size* of the graph.

2 Type A Measures

2.1 Categorization

In these measures, the basic premise is to see if a path exists between two nodes, say u and v , after the removal of nodes. We are not particularly concerned about the length of the path.

Based on attack, that is constraints on the vertices that need to be removed *region-based connectivity* [16]

Based on damage – quantification of change in structure – after the removal of vertices (attacks)

- number of components (connectivity, toughness)
- size of components (integrity, variants of integrity)
-

(in terms of damage, average case vs extremal case)

Structural change	Measures
No. of components	vertex-connectivity, toughness,
Order of components	integrity, fragmentability,
No. and order of components	tenacity, critical node, average connectivity,

2.2 Definitions

2.2.1 Based on the Number of Components

Vertex connectivity ($\kappa(G)$) – A graph is k -connected if the number of vertices that need to be removed to have a residual graph with at least two components is greater than k . The maximum value of k for which the graph is k -connected is the *vertex connectivity*, κ .

Toughness quantifies ...

Toughness ($\tau(G)$) – A graph is t -tough if the number of vertices that need to be removed to have a residual graph with $\omega \geq 2$ components is at least $t\omega$. The maximum value of t for which the graph is t -tough is the

toughness, τ of G . Equivalently, if $S \subset V$ is a cut set, then the *toughness* of G is

$$\tau(G) = \min_{S \subset V} \{ |S| / \omega(G') \} \quad (1)$$

Vertex integrity, or simply integrity of a graph quantifies the possibility of decomposing a network into smaller sized components by removing a small number of nodes.

2.2.2 Based on the Order of Components

Integrity ($\mathcal{I}(G)$) – Let $S \subset V$ be the subset of nodes removed from G , and $m(G')$ be the order of the largest component of the residual graph G' , then the *integrity* of G is

$$\mathcal{I}(G) = \min_{S \subset V} \{ |S| + m(G') \} \quad (2)$$

Mean integrity [17], weak integrity [1], neighbour-integrity [18].

Fragmentability $\mathcal{F}(G)$ –

2.2.3 Based on the Number and Order of Components

Tenacity $\mathcal{T}(G)$ –

Average connectivity ($\bar{\kappa}(G)$) –

Critical node

2.3 Computational Complexity

3 Applications

4 Ways to Improve Structural Stability

- By adding more links – not very suitable in many practical scenarios, as adding links is not always feasible due to economic reasons or practical issues. Moreover, adding more links also increase the potential attack surface for the attackers.

- Here, we propose an alternative approach that does not involve strategic addition of edges or links to improve network structural properties. The basic idea is to ensure the availability and operational integrity of a very small subset of nodes and edges at all times by protecting them from failures/attacks. We call such a subset of nodes and edges *trusted*. They correspond to the network components that are more reliable owing to more resources and higher security measures. For instance, in terms of protection against physical attacks, defense mechanisms against jamming, protected memory, and sophisticated authentication mechanisms. We then show that the overall network connectivity and robustness can be significantly improved by making a very small subset of nodes and edges trusted, even if the original network is sparse. Moreover, by controlling the number of trusted nodes, any desired level of connectivity could be obtained. Thus, instead of the idea of *redundancy* to improve structural properties of networks, we exploit the notion of *reliability and defense* (trustedness) of a small subnetwork to improve network structure.

References

- [1] A. Kirlangic, "On the weak-integrity of graphs," *Journal of Mathematical Modelling and Algorithms*, vol. 2, no. 2, pp. 81–95, 2003.
- [2] C. A. Barefoot, R. Entringer, and H. Swart, "Vulnerability in graphs – A comparative survey," *J. Combin. Math. Combin. Comput*, vol. 1, no. 38, pp. 13–22, 1987.
- [3] W. Goddard, "Measures of vulnerability—the integrity family," *Networks*, vol. 24, no. 4, pp. 207–213, 1994.
- [4] D. Moazzami, "Vulnerability in graphs—a comparative survey," *Journal of Combinatorial Mathematics and Combinatorial Computing*, vol. 30, pp. 23–32, 1999.
- [5] F. Harary, "The maximum connectivity of a graph," *Proceedings of the National Academy of Sciences*, vol. 48, no. 7, pp. 1142–1146, 1962.
- [6] D. Bauer, H. Broersma, and E. Schmeichel, "Toughness in graphs – A survey," *Graphs and Combinatorics*, vol. 22, no. 1, pp. 1–35, 2006.
- [7] M. Cozzens, D. Moazzami, and S. Stueckle, "The tenacity of a graph," *Graph theory, Combinatorics, and Algorithms*, vol. 1, p. 2, 1995.
- [8] K. S. Bagga, L. W. Beineke, W. Goddard, M. J. Lipman, and R. E. Pippert, "A survey of integrity," *Discrete Applied Mathematics*, vol. 37, pp. 13–28, 1992.
- [9] D. Woodall, "The binding number of a graph and its anderson number," *Journal of Combinatorial Theory, Series B*, vol. 15, no. 3, pp. 225–255, 1973.
- [10] K. Edwards and G. Farr, "Fragmentability of graphs," *Journal of Combinatorial Theory, Series B*, vol. 82, no. 1, pp. 30–37, 2001.
- [11] L. W. Beineke, O. R. Oellermann, and R. E. Pippert, "The average connectivity of a graph," *Discrete mathematics*, vol. 252, no. 1-3, pp. 31–45, 2002.
- [12] B. Mohar, "Isoperimetric numbers of graphs," *Journal of Combinatorial Theory, Series B*, vol. 47, no. 3, pp. 274–291, 1989.

- [13] S. Hoory, N. Linial, and A. Wigderson, "Expander graphs and their applications," *Bulletin of the American Mathematical Society*, vol. 43, no. 4, pp. 439–561, 2006.
- [14] H. J. LeBlanc, H. Zhang, X. Koutsoukos, and S. Sundaram, "Resilient asymptotic consensus in robust networks," *IEEE Journal on Selected Areas in Communications*, vol. 31, no. 4, pp. 766–781, 2013.
- [15] J. Wu, M. Barahona, Y.-J. Tan, and H.-Z. Deng, "Spectral measure of structural robustness in complex networks," *IEEE Transactions on Systems, Man, and Cybernetics-Part A: Systems and Humans*, vol. 41, no. 6, pp. 1244–1252, 2011.
- [16] A. Sen, B. H. Shen, L. Zhou, and B. Hao, "Fault-tolerance in sensor networks," in *INFOCOM 2006: 25th IEEE International Conference on Computer Communications*, 2006.
- [17] G. Chartrand, S. Kappor, T. McKee, and O. Oellermann, "The mean integrity of a graph," in *Recent Studies in Graph Theory*, K. VR, Ed. Vishwa International Publications, 1989, pp. 70–80.
- [18] V. Aytacı, "Vulnerability in graphs: the neighbour-integrity of line graphs," *International Journal of Computer Mathematics*, vol. 82, no. 1, pp. 35–40, 2005.

Appendix B

Associate Editor Comments to Author (Dr Jon Crowcroft):

thank you for the submission - as you can see the reviewers have some suggestions for revisions - i think these are really minor, so please follow what they suggest, and well done!

Reviewer: 1:

The paper introduces the concept of intermediacy, a metric to identify major publications (knowledge) that have led from one paper to another. Overall, I enjoyed reading this work - the text was concise, and I appreciated the clear explanations and use of figures. I also think that intermediacy as a concept has value. There were a few locations where I think the text requires clarification, but I believe that is easy to fix. My main comments apply to most papers that present a new metric: it's important to clearly state the need for it, and how it complements existing metrics. Currently the paper briefly dismisses main path analysis claiming that the results are counter-intuitive (and therefore it is necessary to develop a new metric). However, the paper lacks a rigorous literature review and, IMHO, this is the wrong strategy. It would be better to provide a meaningful discussion of the metrics available, highlighting their strength and weaknesses when used in different forms of analysis. This should then naturally lead to a 'gap in the market' which intermediacy fills. Of course, you should then highlight the strength/weaknesses of intermediacy, explaining how future researchers should interpret the results if they use it. In the below text I have tried to provide more specific comments that could help in this goal. But congratulations on a nice piece of work, and I hope the feedback can be of use.

- I like the introduction - it is clear and concise. However, it would be good if you could add some further motivational discussion. You mention main path analysis, and then immediately present your approach. Yet, it is not clear why your approach is required, i.e. what are the limitations that you seek to address? A few sentences describing the problems with prior main path analysis techniques would be useful for the reader.

Thank you. This is a good point. We have improved the motivation for our research provided in the introduction.

- "we will show that main path analysis tends to favor longer citation paths over shorter ones, whereas intermediacy has the opposite tendency" - again, it would be useful to add some further context here: what is the benefit of your technique favouring shorter paths? You quickly dismiss main path analysis as counter-intuitive, but it would be better if you could provide a more rigorous analysis. Otherwise, it is not clear to a reader why it is necessary to develop the intermediacy metric, and what analytical benefits it brings. Just to be clear, I *do* think there is a benefit; my suggestion is simply to make this explicit within the early parts of the paper. A positive step would be to add a Background section to describe (and critique) past metrics/studies.

As mentioned above, based on your comments, we have made various improvements to the introduction. However, we think it is difficult to be more rigorous or more explicit in the introduction. This would make the discussion in the introduction quite complex. We prefer to keep the discussion in the introduction simple.

- "we argue that intermediacy yields better results than main path analysis". To address the above comment, you could expand on this assertion. Why do you argue this - could you briefly summarise your evidence here? I would also mention that (hopefully) other scientometrics researchers will use

intermediacy in their studies. Making a strong case for what extra value it brings to a bibliometric study would help in this.

We need the entire paper to present our evidence and to make a strong case for intermediacy. We don't see how all of this can be done in the introduction.

- Minor: I would suggest making the definition of "active edge" more explicit. Definition 2.2 covers what an "active path" is, but this relies on the assumption that the reader understands an "active edge". After reading further on, this obviously becomes clear. But it would be useful to have a brief explicit definition before definition 2.2.

We have made the definition of active and inactive edges somewhat more explicit.

- When I started reading Section 2(B), I assumed that this would offer guidelines for setting ρ . However, the conclusion of "one can set the probability p to a value that one considers appropriate for a particular analysis" does not really answer the more meaningful question that the section poses: how should I select an appropriate ρ for my particular analysis? You later use $1/k$ - is this your general recommendation?

No, we don't have a general recommendation. The user may choose a value for p based on his or her subjective preferences. Figure 1 could support the user in choosing a value for p . In our view, there is no optimal value for p . Different values of p offer different perspectives on the data. Of course, the user could try out different values for p . This may offer deeper insights into the data.

- It is clear what the implications of changing ρ are, but setting it seems to require the researcher to know what path properties they are looking for in their citation graph (i.e. shortest path or num independent paths). Do you have any recommendations or insight here, because selecting a "wrong" ρ would obviously impact the outcome. IMHO, this query could be addressed in one of two ways. Either (1) You could provide a methodology or (approximate) guideline regarding how people could select ρ , or (2) You could give some clear insight into how to interpret results based on the value of ρ . The latter could be done in a table, which summarises implications for certain ranges of ρ . Again, if the answer is $1/k$, you could state this here.

See our response to the previous comment.

- Minor: p6, line 28,48; p10,line 15,23,34; p11, line 33,35,38: table/fig need uppercase letters (you switch between upper and lower case)

We have used the same format as some of recently published papers by the journal that use "Figure" only at the beginning of a sentence. If this is not correct, this can be easily corrected in the final proofs of the paper prepared by the journal staff.

- Page 7, "For the purpose of identifying publications that play an important role in connecting an older and a more recent publication, we consider this behavior to be undesirable". I found this

statement a bit confusing. There are a few statements critiquing main path analysis, but the text tends to be brief and dismissive. Incidentally, framing things as being "undesirable" might be the wrong tactic - it would be better to highlight the different insights that can be gleaned from the two (complementary?) approaches. If you really think main path analysis is scientifically wrong, you should provide clear evidence for this.

To strengthen the evidence against main path analysis, we have extended case study 1. This case study now includes a comparison between intermediacy and main path analysis. Based on our expert knowledge of the literature analyzed in case study 1, we believe that the comparison offers clear evidence against main path analysis.

- With the above in mind, it would be good to compare your results directly against main path analysis. The second case study uses [22], but it isn't mentioned in the first case study. You could then use this to expand your claims. You could explicitly state how the results differ, and the complementary (or superior?) nature of the insights gleaned. As I said before, I don't think you necessarily need to criticise main path analysis - instead, you could just more clearly highlight the value of using *both* techniques within a bibliometric study.

On this point, our perspective is different from the reviewer's. We don't see the value of main path analysis, and therefore we want to explicitly criticize the technique. In our opinion, main path analysis lacks a proper conceptual foundation. It is unclear how the results of main path analysis can be interpreted. We don't feel comfortable presenting intermediacy and main path analysis as complementary techniques.

- I was curious why you limited yourselves to these two specific case studies with single pairs of paper? Could some details be added here? At first, I thought it was because you had the domain expertise to evaluate the correctness of the outcome. However, in Case Study 1 you just briefly discuss the papers, and in Case Study 2 you say that you don't have the expertise to comment on it. It would be valuable to actually try to quantify the correctness of your results - for instance, by performing a survey of relevant researchers to judge accuracy. I wouldn't expect this for this paper, but might be interesting future work.

Thank you for this suggestion. In the revised paper, we briefly mention this as a direction for future research.

- Again, without wishing to strawman the paper, why did you choose not to scale-out your empirical analysis? For instance, you could start with a domain (community detection) and then try many different $\langle s, d \rangle$ pairs, so that you can explore more generalisable trends. When you only present 2 specific case studies, it naturally raises questions related to how applicable this might be to other domains (or even other pairs of papers). Is this a fair comment? If not, it would be good to briefly explain why such an approach is unsuitable for your goals.

Presenting more extensive case studies would be a possibility. However, we prefer to leave this for future research. We believe that our paper makes a strong case for intermediacy. Of course, we haven't fully addressed all questions one could have about intermediacy, but the remaining questions probably can best be addressed in follow-up studies, either by ourselves or by others.

- I liked the short discussion about how intermediacy differs from citation rates in some cases. More generally, I think these discussions are valuable (in any of paper which introduces a new metric). I would recommend expanding this type of discussion, such that you better contrast intermediacy with other relevant metrics to clearly state why people working in scientometrics should use it in the future. For example, you could add a table that summarises the strength/weaknesses of several metrics (citation count etc) including intermediacy.

We are interested in intermediacy for the specific purpose of identifying important paths in citation networks. Other metrics have not been developed for this purpose. Therefore we don't think it would make sense to present a table in which intermediacy is compared with other metrics. Intermediacy serves a different purpose than other metrics, so there is no clear reason to compare them with each other.

In the introduction, we have added a paragraph explaining why intermediacy is different from other metrics, in particular from centrality metrics.

- Quick question: I wasn't clear why you felt intermediacy scores were more useful from an ordinal perspective? Indeed, Spearman often shows stronger correlations than Pearson. But is your ordinal statement simply driven by the the higher correlations you observe, or was there something more fundamental?

We are not interested in correlations being high or low. The reason why we believe that intermediacy scores can best be treated as ordinal values is that it is unclear what it means to say that the intermediacy of one publications is, say, twice as high as the intermediacy of another publication. On the other hand, it is clear what it means to say that the intermediacy of one publication is higher than the intermediacy of another publication.

Also, we use intermediacy scores as a tool to identify important paths. For this purpose, only the ranking of the intermediacy scores of publications matters. The exact scores are not important. The paths identified using intermediacy remain the same when a monotonic transformation is applied to the intermediacy scores of publications.

- Minor: Could you add the X labels for the heatmaps, it would just make it easier to read.

The labels are too long to be added to the horizontal axes of heatmaps without making the resulting figures wider than the page width. We have therefore decided to omit the labels and we hope that the figures are still clear to read.

- Minor: p10,line 12: character error.

Thanks. We have fixed this.

Reviewer: 2:

This is an interesting well-written paper that seeks to trace the historical development of scientific knowledge. The authors show that the proposed intermediacy measure outperforms main path analysis – a known approach for tracing historical developments in citation networks. I have the following major concerns:

1) Several measures exist in graph theory literature such as betweenness centrality, closeness etc. that are similar to the proposed intermediacy measure. I would like authors to justify why these known measures are not suited in this context? also, justify the proposal for this new measure.

We have added a paragraph at the end of the introductory section to address this comment.

2) The authors claim that the intermediacy increases if a given path is contracted or extend. However, the authors fail to give the practical implication of this phenomenon in a citation network, which is a fixed graph.

We have added a few sentences in Section 2 to clarify this issue.

3) Another well-known measure that can measure the quality and strength of links is Kirchhoff index, I would ask authors to show that the intermediacy measure outperforms Kirchhoff index in capturing the historical development.

Thank you for pointing out the possible connection between our work and the Kirchhoff index. Triggered by your comment, we have studied the relevant literature. In particular, we have studied the literature on the notion of resistance in networks. In Section 2(d) in our paper, we have added a discussion of the notion of resistance. We provide a theoretical argument for preferring intermediacy over a similar approach based on resistance. More specifically, we show that, unlike intermediacy, resistance does not satisfy our path addition property.

4) The manuscript contains trivial statements which are written as detailed theorems in the text, and this can be avoided by merely citing the relevant literature.

We don't understand this comment. It is not clear to us which 'trivial statements' and which 'relevant literature' the reviewer is referring to.

5) Attached find a white paper that has been compiled for a relevant project, containing several classic approaches to do what authors are trying with intermediacy measure. I would ask the authors to justify mathematically and/or empirically how the proposed measure is comparable with existing indices.

Thank you for sharing this white paper with us. The white paper is an unfinished draft. It is therefore hard for us to compare our proposed ideas with the ideas discussed in this white paper.